# Flavone and Hydroxyflavones Are Ligands That Bind the Orphan Nuclear Receptor 4A1 (NR4A1)

**DOI:** 10.3390/ijms24098152

**Published:** 2023-05-02

**Authors:** Miok Lee, Srijana Upadhyay, Fuada Mariyam, Greg Martin, Amanuel Hailemariam, Kyongbum Lee, Arul Jayaraman, Robert S. Chapkin, Syng-Ook Lee, Stephen Safe

**Affiliations:** 1Department of Veterinary Physiology and Pharmacology, Texas A&M University, College Station, TX 77843, USA; mi-ok.lee@ag.tamu.edu (M.L.); supadhyay@cvm.tamu.edu (S.U.); fuada@tamu.edu (F.M.); gmartin@tamu.edu (G.M.); ahailemariam@tamu.edu (A.H.); 2Department of Biochemistry and Biophysics, Texas A&M University, College Station, TX 77843, USA; 3Department of Chemical and Biological Engineering, Tufts University, Medford, MA 02155, USA; kyongbum.lee@tufts.edu; 4Artie McFerrin Department of Chemical Engineering, Texas A&M University, College Station, TX 77843, USA; arulj@mail.che.tamu.edu; 5Department of Nutrition, Texas A&M University, College Station, TX 77843, USA; robert.chapkin@ag.tamu.edu; 6Department of Food Science and Technology, Keimyung University, Daegu 42601, Republic of Korea; syng-ook.lee@kmu.ac.kr

**Keywords:** flavone, hydroxyflavones, binding, NR4A1

## Abstract

It was recently reported that the hydroxyflavones quercetin and kaempferol bind the orphan nuclear receptor 4A1 (NR4A1, Nur77) and act as antagonists in cancer cells and tumors, and they inhibit pro-oncogenic NR4A1-regulated genes and pathways. In this study, we investigated the interactions of flavone, six hydroxyflavones, seven dihydroxyflavones, three trihydroxyflavones, two tetrahydroxyflavones, and one pentahydroxyflavone with the ligand-binding domain (LBD) of NR4A1 using direct-binding fluorescence and an isothermal titration calorimetry (ITC) assays. Flavone and the hydroxyflavones bound NR4A1, and their K_D_ values ranged from 0.36 µM for 3,5,7-trihydroxyflavone (galangin) to 45.8 µM for 3′-hydroxyflavone. K_D_ values determined using ITC and K_D_ values for most (15/20) of the hydroxyflavones were decreased compared to those obtained using the fluorescence assay. The results of binding, transactivation and receptor–ligand modeling assays showed that K_D_ values, transactivation data and docking scores for these compounds are highly variable with respect to the number and position of the hydroxyl groups on the flavone backbone structure, suggesting that hydroxyflavones are selective NR4A1 modulators. Nevertheless, the data show that hydroxyflavone-based neutraceuticals are NR4A1 ligands and that some of these compounds can now be repurposed and used to target sub-populations of patients that overexpress NR4A1.

## 1. Introduction

Flavonoids are polyphenolic phytochemicals produced in fruits, nuts, and vegetables, and these compounds share a common phenylchromene-4-one structure substituted with one or more hydroxyl group substituents [1,2,3]. Dietary consumption of flavone and flavonoid-containing foods is associated with improved health benefits which have been linked to their antioxidant and anti-inflammatory activities and their direct effects on other signaling pathways [4,5,6,7,8,9]. For example, in the Framingham offspring cohort, “higher long term dietary intakes of flavonoids” and related compounds are associated with a lower risk of Alzheimer’s disease and related dementias [10]. In another study, higher dietary intake of flavonoids was associated with lower rates of obesity [11] and this association was also observed for other health benefits including increased lifespan [12,13]. Flavonoids have also been used in pre-clinical animal models and in cell culture studies to investigate their chemotherapeutic effects for treating multiple diseases including cancer and non-cancer endpoints such as endometriosis, intestinal inflammation, and neuronal diseases [14,15,16,17,18,19,20,21,22,23,24]. Although the mechanisms of action of flavonoids are complex, in some studies, major contributing flavonoid-induced pathways were identified. For example, several flavonoids interact with the aryl hydrocarbon receptor (AhR), and both cardamonin and baicalein are polyhydroxylated flavonoids that exhibit AhR-dependent inhibition of intestinal inflammatory activity in rodent models [24,25,26,27]. Flavonoids also bind other receptors, enzymes, and multiple proteins; however, the mechanisms associated with their chemopreventive, and chemotherapeutic effects are not well-defined [28].

Many neutraceuticals are flavonoid-based products and traditional medicines containing these compounds represent a multibillion-dollar component of the pharmaceutical industry. Although preclinical laboratory studies on flavonoids are promising, the results of human clinical trials with flavones and other flavonoids have been disappointing [24,28]. Moreover, there are major problems with using flavonoids as pharmaceuticals in clinical trials due to their rapid metabolism and poor uptake caused by unfavorable pharmacokinetics/pharmacodynamics. Another problem that contributes to the relative ineffectiveness of flavonoids in clinical trials is the lack of precision in using “flavonoid-based” pharmaceuticals since the cellular targets for flavonoids in patient populations are not well-defined.

Studies in this laboratory have been focused on the orphan nuclear receptors NR4A1 (Nur77) and NR4A2 (Nurr1), which are transiently induced by diverse stressors in normal cells and overexpressed in many stress-related diseases including solid tumors [29,30,31]. In solid tumors, NR4A1 and NR4A2 are negative prognostic factors, they exhibit pro-oncogenic activities, and these can be inhibited by bis-indole derived compounds (CDIMs) which bind NR4A1 and NR4A2 and act as antagonists in cancer cells [31,32]. These CDIM compounds have minimal effects on NR4A3, and this receptor has not been extensively investigated in solid tumors [31]. Many of the anticancer activities of flavonoids and other polyphenolics resemble those reported for CDIM/NR4A1/2 antagonists and like the antagonists, the flavonoids quercetin and kaempferol also bound NR4A1 and exhibited NR4A1 antagonist activities in rhabdomyosarcoma cells and inhibited tumor growth in an athymic mouse xenograft model [33]. These results observed in Rh30 cells prompted the study reported herein where we show that flavone and several hydroxyl-substituted flavones bind NR4A1 and confirm that this important class of polyphenolics are NR4A1 ligands. The results complement our recent studies on quercetin and kaempferol; however, our structure–NR4A1 binding, K_D_ values and structure-dependent transactivation results are poorly correlated. The results suggest that the hydroxyflavones are selective NR4A1 modulators and their agonist or antagonist activities are cell- and gene-context dependent and require individual compound studies on their mode of action and efficacy.

## 2. Results

Recent studies show that polyphenolic compounds such as quercetin, kaempferol, broussochalcone and resveratrol bind the orphan nuclear receptor NR4A1, and in cancer cell models these compounds act as NR4A1 antagonists [33,34,35]. In this study, we examined the direct binding of flavone and structurally diverse hydroxyflavones which contained a variable number of hydroxyl groups with different substitution patterns to the ligand-binding domain of NR4A1. We initially used a fluorescent binding assay that measured the quenching of the fluorescence of a Trp residue in the LBD of NR4A1. The binding of selected hydroxyflavones to the LBD of NR4A1 and the calculated K_D_ values are shown in Figure 1 and values for all 20 compounds are summarized in Table 1. The fluorescent Trp quenching assay was used to derive the direct binding curve for the 3-, 5-, 6-, 7-, 3′- and 4′-hydroxyflavones. 3-Hydroxyflavone was insoluble in the fluorescence assay and results were inconsistent. The K_D_ values for the remaining compounds varied from 45.8 to 1.4 µM for 3′- and 5-hydroxyflavone, respectively. Flavone also bound NR4A1, and the K_D_ value was 3.4 µM. We also investigated the direct binding of several isomeric dihydroxyflavones to the LBD of NR4A1, and the K_D_ values varied from 21.4 µM for 7,3′-dihydroxyflavone to 0.66 µM for 3,6-dihydroxyflavone. There were several interesting changes in the binding affinities between the parent mono-hydroxyflavones and their corresponding dihydroxyflavone analogs where the K_D_ values for most of the dihydroxyflavones were lower than those of the corresponding mono-hydroxyflavones. For example, K_D_ values for 3′- and 4′-hydroxyflavone were 45.8 µM and 13.0 µM, respectively whereas the K_D_ for 3′,4′-dihydroxyflavone was 0.96 µM. We also examined K_D_ values for the binding of several tri-, tetra- and penta-hydroxyflavones to NR4A1 and they varied from 0.36 µM for 3,5,7-trihydroxyflavone (galangin) to 1.85 µM for 5,6,7-trihydroxflavone. Inspection of the binding data for the tri-, tetra- and penta-hydroxyflavones showed a range of K_D_ values that did not indicate any specific structure–activity relationships among these compounds. Flavones substituted with hydroxyl groups at the 3,5-, 5,7- and 3,5,7-positions in the flavone backbone tended to bind with higher affinity to NR4A1 than their corresponding positional isomers did. However, based on the results summarized in Table 1, the structure–binding relationships between hydroxyflavones and NR4A1 were not apparent.

Since the K_D_ values obtained using the fluorescence binding assay were highly variable and did not exhibit any consistent structure-dependent effects, we also used the isothermal titration calorimetry (ITC) assay which measures the heat lost or gained due to ligand–NR4A1 binding and this assay provides both K_D_ and ΔG values associated with these interactions. The results (Table 1) show that K_D_ values were highly variable (444 to 0.001 µM) and did not correlate with the K_D_s obtained using the fluorescence assay. Fifteen of the twenty hydroxyflavones exhibited ITC-derived K_D_ values lower than those observed using the fluorescence assay whereas the inverse was true for five of these compounds. Figure 2 illustrates the binding curves and thermodynamic data for 6- and 7-hydroxyflavone. It was noteworthy that some of the K_D_ values using the ITC assay were in the low nM range including those for 5,7-dihydroxyflavone (K_D_ = 1 nM) and 3,5,7-trihydroxyflavone (K_D_ = 1 nM), and there was rank order correlation among the hydroxyflavones between their K_D_ and ΔG values as determined by the ITC binding assay. There was no obvious explanation for the in vitro assay-dependent differences in K_D_ values from the two binding assays except that the ITC assay integrated ligand binding not only within the ligand-binding domain but also on other surfaces of the receptor.

We also examined the effects of flavone and the hydroxyflavones on NR4A1-dependent transactivation in Panc1 pancreatic cancer cells transfected with GAL4-NR4A1/UAS-luc constructs (Figure 3). Cells were treated with 25 or 50 µM concentrations of the flavones and the effects on luciferase activity were variable; 4′-hydroxy-, 6-hydroxy-, 7-hydroxy-, 5,6,7-trihydroxy-, 5,7,3′,4′-tetrahydroxy- and 3,5,7,3′,4′-pentahydroxyflavone (quercetin) significantly induced luciferase activity. In contrast, only 3,5,7-trihydroxyflavone decreased luciferase activity; the remaining hydroxyflavones increased or decreased luciferase activity but these responses were not significant. We observed minimal toxicity using these flavone concentrations (<20% floating cells) and this was consistent with our previous studies on their AhR-dependent activities [36,37]. These results contrasted those of previous studies using quercetin (3,5,7,3;4′-pentahydroxyflavone) and kaempferol (3,4′,5,7-tetrahydroxyflavone), which decreased luciferase activity in Rh30 rhabdomyosarcoma cells transfected with the same constructs [33]; in contrast, the results obtained in Panc1 cells were highly variable. It is possible that higher concentrations (>50 µM) are required to elicit induced or inhibited transactivation; however, these higher concentrations were not used due to the toxicities of these compounds at higher concentrations.

Based on these variable results from Panc1 cells, we repeated the NR4A1-dependent transactivation assays in Rh30 cells using a high concentration (50 µM) that was not cytotoxic. Both kaempferol and quercetin decreased transactivation as previously reported in this cell line [33] and similar results were observed for 1,1-bis(3′-indolyl)-1-(3,5-dichlorophenyl)methane (DIM-3,5-CI_2_) which is a highly potent bis-indole-derived NR4A1 antagonist [32] that was used as a control for this assay. There were more distinct differences between the hydroxyflavones that inhibited transactivation in Rh30 cells compared to those observed in Panc1 cells and potent inhibition was observed for 4′-hydroxy, 3,6-dihydroxy, 3,7-dihydroxy, 3,5,7-trihydoxy and 3,5,3′,4-tetrahydroxyflavone (Figure 4). Moreover, 6-hydroxy, 5,6,7-trihydroxy and 5,7,4′-trihydroxyflavone also decreased transactivation in Rh30 cells whereas the remaining compounds did not increase or slightly increased transactivation in Rh30 cells. Interestingly, 7-hydroxy and 3′,4′-dihydroxyflavone induced (<2.4-fold) luciferase activity in both cell lines. The results obtained for Rh30 cells were more definitive than those observed for Panc1 cells in terms of identifying hydroxyflavones that inhibited NR4A1-dependent transactivation, but structure–activity relationships were not observed.

We also carried out modeling studies using Maestro/Schrodinger software on the interaction of flavones/hydroxyflavones withTMY301 and TMY302 sites in the LBD of NR4A1 (3V3Q) [38]. The docking scores for interactions of hydroxyflavones at site TMY301 and TMY302 are summarized in Table 2 and Appendix A. The results show that, with the exception of 3,6-dihydroxyflavone, the docking scores indicated that the remaining hydroxyflavones more favorably interacted with site TMY301 than they did with site TMY302. The variation in docking scores for site TMY301 ranged from –7.059 (5,7,3′,4′-tetrahydroxyflavone) to –4.695 (3,7-dihydroxyflavone) and those for site TMY302 ranged from –5.85 (3,7-dihydroxyflavone) to –4.582 (flavone). Figure 5 illustrates the interactions of the compounds that bind with the highest and lowest affinities to TMY301 and TMY302 in the LBD of NR4A1. Previous studies show that ethyl 2-[2,3,4-trimethoxy-6-(octanoyl)phenyl] acetate also binds both sites, which are close to the surface of the ligand pocket, and the docking scores for these sites are –4.83 and –5.62, respectively, showing a preference for TMY302. The results indicate that flavone and the hydroxyflavones interact with both sites in the LBD of NR4A1 with preferential binding to TMY301 (exception: 3,6 -dihydroxyflavone). An examination of the docking scores, K_D_ values for two assays and -ΔG values did not show any obvious structure-binding or structure–docking relationships and this was consistent with the lack of structure–activity (transactivation) correlations. Many of the early studies on high-affinity ligands such as steroid hormones and their binding to corresponding cognate receptors exhibit structure–activity relationships. However, subsequent development of lower-affinity receptor ligands for nuclear receptors such as the estrogen receptor α (ERα) do not necessarily exhibit structure–activity relationships; this is illustrated by selective ER modulators (SERMs) which have and are being developed for hormonal therapies [39,40]. Thus, results of this study suggest that hydroxyflavones may also be selective NR4A1 modulators and this is further discussed below.

## 3. Discussion

The orphan nuclear receptors NR4A1, NR4A2 and NR4A3 are overexpressed in multiple diseases including solid tumors [31] where both NR4A1 and NR4A2 act as pro-oncogenic factors. In contrast, based on the results of NRA41 and NR4A3 knockout studies on mice, it was shown that the dual knockdown of both NR4A1 and NR4A3 in mice resulted in the development of leukemia, and NR4A1 and NR4A3 exhibited tumor suppressor-like activities in most blood-derived tumors [41,42]. Many studies have focused on the role and functions of orphan NR4A and there is now a growing body of literature on the identification and functions of these receptors, although endogenous NR4A ligands have not been identified. Compounds that bind NR4A1 and NR4A2 have been reported whereas less is known about ligands that bind NR4A3 [29,30,31]. Interestingly, many of the initial compounds identified as NR4A1 or NR4A2 ligands are natural products and some of them serve as scaffolds for the synthesis of more potent analogs [43,44]. For example, phytochemicals/microbial metabolites that bind NR4A1 include cytosporone B, celastrol, fangchinoline, isoalantolactone, polyunsaturated fatty acids, a bacterial bile acid metabolite, quercetin and kaempferol, a sponge-derived sesterterpenoid 12-deacetyl-12-epi-scalaradial and prostaglandin A2 [33,34,35,43,45,46,47,48,49,50,51]. NR4A2 also binds natural products including prostaglandins E1 and A1, 5,6-dihydroxyindole (a dopamine metabolite) and unsaturated fatty acids [49,50,51,52,53,54,55]. Interestingly the only compounds reported to bind both NRA41 and NR4A2 are the unsaturated fatty acids and arachidonic acid and docosahexaenoic acid bind both receptors [51,54].

Kaempferol and quercetin are ligands that bind NR4A1 and exhibit NR4A1 antagonist activities in Rh30 rhabdomyosarcoma cells [33] and this has previously been observed for other NR4A1 antagonist such as the bis-indole derived compounds that also bind NR4A1 [31]. Since many flavonoids exhibit anticancer activities [23,56,57] similar to those observed with kaempferol and quercetin, we further investigated flavone and hydroxyflavones as potential ligands for NR4A1. Many of these flavonoids are extensively used as neutraceuticals and traditional medicines worldwide; however, their application as precision medicines has been limited since their mechanisms of action and specific intracellular targets are not fully understood or identified. We initially examined these compounds as ligands for NR4A1 using a direct binding assay which measures the loss of fluorescence associated with a tryptophan residue in the ligand-binding domain of the receptor [33]. Flavone itself bound NR4A1 with a K_D_ of 3.4 µM, and among the hydroxyflavone isomers examined only 5-hydroxyflavone exhibited a lower K_D_ (1.4 µM) than flavone did and 3′-hydroxyflavone exhibited the lowest binding affinity with a K_D_ value of 45.8 µM. Binding studies with 3-hydroxyflavone gave unreliable results due to the aqueous insolubility of this compound. Subsequent binding of hydroxyflavones to NR4A1 and their corresponding K_D_ values were highly variable among several di-pentahydroxy flavones and ranged from 0.36 µM for 3,5,7-trihydroxyflavone (galangin) to 16.4 µM for 3,4,7′-trihydroxyflavone. K_D_ and ΔG values were also determined for the hydroxyflavones using the ITC binding assay and for 15/20 of these compounds, lower K_D_ values were observed in the ITC vs. the fluorescence binding assay (Table 1). Although the K_D_ and ΔG values were correlated in the ITC assay, there were no obvious structure–binding/activity relationships using this assay and correlations between the two different sets of K_D_ values for the hydroxyflavones were not observed.

Molecular modeling studies using Maestro/Schrodinger software investigated interactions of the hydroxyflavones with amino acid side chains in the LBD of NR4A1 (Table 2), and Figure 5 illustrates the interactions of 5,7,3′,4′-tetrahydroxyflavone compared to those of 3,6-dihydroxyflavone in the NR4A1-TMY301 binding site and the interactions of 3,7-dihdroxyflavone and flavone in the NR4A1-TMY302 binding site. These compounds have the lowest and highest docking values for each binding site; for each binding site there is considerable overlap in their interactions with common amino acid side chains but also differences which presumably dictate the differences in their docking scores. It is also apparent that different sets of amino acids are important for the interactions of the flavonoids with the TMY301 and TMY302 binding sites. However, there was also no correlation between the K_D_ values and docking scores for the compounds illustrated in Figure 5 and other hydroxyflavones listed in Table 2. Thus, the binding and modeling studies do not identify structure–activity relationships that correlate with K_D_ values observed in the direct fluorescence and ITC binding assays.

We also examined the effects of the flavones on NR4A41-dependent transactivation using a GAL4-NR4A1 chimera and UAS-luc constructs and observed that ligand dependency increased, decreased and had no effect on luciferase activity. Previous studies showed that quercetin and kaempferol acted as NR4A1 antagonists in Rh30 rhabdomyosarcoma cells [33] and decreased NR4A1-dependent transactivation in this cell line. Our initial studies showed some variation in the effects of hydroxyflavonoids on NR4A1-dependent luciferase activity in Panc1 cells which appeared to be relatively resistant to the effects of hydroxyflavones as either agonists or antagonists using the GAL4/UAS-luc assay. In contrast, several hydroxyflavones were NR4A1 antagonists in Rh30 cells and this was consistent with previous studies with quercetin and kaempferol which also exhibited functional activity as NR4A1 antagonists in Rh30 cells [33]. This suggests that hydroxyflavones may be selective NR4A1 modulators and that the effects of individual compounds may be cell context-, response- and gene-specific. This selectivity is observed for many nuclear receptor ligands and their receptors [39,40,58]. For example, selective estrogen receptor modulators (SERMs) such as fasoldex, tamoxifen and raloxifene exhibit different K_D_ values for ER binding, and among structurally diverse estrogenic compounds their K_D_ values and response selectivities are also highly variable [39].

Thus, our results did not show any structure–K_D_ (binding), structure–activity (transactivation), or structure–docking relationships for twenty flavonoids using two complementary binding assays and two transactivation assays and by modeling interactions with NR4A1. The addition of more flavones may be needed to resolve some of these issues; however, examinations of previous structure–binding and structure–activity relationships of flavonoids with other receptors gave similar results which precluded the development of structure–potency correlations. Studies in this laboratory on structure–activity relationships for hydroxyflavones as AhR agonists and antagonists also gave highly variable results. Although K_D_ values for hydroxyflavone–AhR binding were not determined, the effects of hydroxyflavones as inducers of AhR-responsive CYP1A1, CYP1B1 and UGT1A1 genes were determined in Caco2 cells [36,37]. There were some structure-dependent effects among the disubstituted flavones where most 6-substituted dihydroxyflavones were agonists and 7-substituted dihydroxyflavones were antagonists of CYP1A1 induction. However, both 6- and 7-substituted dihydroxyflavones were AhR agonists for the induction of CYP1B1 [36], and for more highly substituted hydroxyflavones, substituent effects on activity were not observed. The lack of predictive structure–activity relationships of hydroxyflavones as AhR ligands was also summarized in a recent review which clearly demonstrates the cell context- and response-dependent variability of hydroxyflavones as AhR agonists and antagonists [25]. Another study used modeling to determine the docking scores of flavonoids for several receptors. For example, the docking values of 17β-estradiol, 5,7,4′-trihydroxyflavone (apigenin), 5,7,3′,4′-tetrahydroxyflavone (luteolin) and 3,7,3′,4′,5-pentahydroxyflavone (quercetin) for ERα were –10.944, –9.698, –9.311 and –8.911, respectively [59]. Functional studies were not determined; however, another report showed that 5,7,4′-trihydroxyflavone exhibited both ER binding and functional activity whereas quercetin did not bind ERα [60]. It was concluded that “the way in which these molecules exert an agonist or antagonist effect on nuclear receptors is not easily predictable and it likely depends on the specific co-activator/co-repressor population of different tissues” [59]. These results parallel the effects observed for hydroxyflavones in this study, indicating that for NR4A1 the hydroxyflavones are also selective receptor modulators. This implies that clinical applications of specific hydroxyflavones cannot necessarily be derived from structure–binding results but must be individually evaluated and optimized in preclinical studies prior to clinical applications.

## 4. Materials and Methods

### 4.1. Chemicals and Cell Culture

All the hydroxyflavones and flavone were purchased from Indofine Chemical Co. (Hillsbrough, NJ, USA); the purities for a 7,3′,4′-. 3,7,3′-, and 5,7,4′-trihydroxyflavone were 97% and all remaining compounds were 98-99% pure. Cancer cells were maintained in Dulbecco’s modified Eagle’s medium (DMEM) nutrient mixture with Ham’s F-12 (DMEM/F-12; Sigma-Aldrich, St. Louis, MO, USA) supplemented with 5% fetal bovine serum (FBS; Sigma-Aldrich) and a 10 mL/L 100× antibiotic–antimycotic solution (Gibco, Thermo Fisher Scientific, Waltham, MA, USA). Cells were maintained at 37 °C in the presence of 5% CO_2_. The 3,5-disubstituted CDIM was synthesized in our laboratory [32].

### 4.2. Plasmids and Luciferase Assay

The UAS-luciferase reporter gene containing 5 GAL4 binding sites and the GAL4-NR4A1 chimera containing the GAL4 DNA-binding domain which linked to the ligand-binding domain (LBD) of NR4A1 were used in transactivation assays in Panc1 pancreatic cancer cells. Panc-1 cells were plated in a 24-well plate (3 × 104 cells/well) and grown for 24 h. Cells were then co-transfected with each construct (250 ng) and the β-gal expression construct (25 ng) using GeneJuice (Novagen, Madison, WI, USA) in accordance with the manufacture’s protocol. The medium was removed after 18 h and replaced with a 2.5% charcoal-stripped FBS-supplemented medium containing either DMSO or flavonoids (25 and 50 µM) in DMSO. After 24 h, cells were then lysed, and cell extracts were processed to measure luciferase activity. Luciferase activity was normalized by dividing the β-gal expression and each treatment was determined in three independent experiments. Data were expressed as mean ± SE and significant (*p* < 0.05) effects were noted.

### 4.3. Quenching of NR4A1 Tryptophan Fluorescence by Direct Ligand Binding

Tryptophan fluorescence spectra were obtained essentially as described [33]. Briefly, the ligand-binding domain of NR4A1 at the final concentration of 0.5 µmol/L in 1.0 mL of phosphate-buffered saline (PBS, pH 7.4) was used for fluorescence measurements. The protein was incubated for 3 min at 25 °C in a temperature-controlled (Quantum Northwest TC125) fluorescence spectrophotometer (Varian Cary Eclipse). The fluorescence spectra were obtained using an excitation wavelength of 285 nm (excitation slit width = 5 nm) and an emission wavelength range of 300–420 nm (emission slit width = 5 nm). Aliquots (0.1 µL/aliquot) of the ligand (10 mmol ligand/L ethanol) were then added to the cuvette containing NR4A1 to reach a final ligand concentration of up to 60 µmol/L. After each aliquot of the ligand, the NR4A1/ligand solution was incubated at 25 °C for 3 min and the loss of tryptophan fluorescence was measured as described above. The addition of DMSO only (up to a final volume of 3.0 µL) had no effect on NR4A1 tryptophan fluorescence. Ligand binding to NR4A1 was determined by measuring the NR4A1 tryptophan fluorescence intensity at the emission wavelength of 330 nm and the resulting data were used to calculate K_D_ values. Fluorescence intensity at each ligand concentration was used to correct the NR4A1 tryptophan fluorescence intensity as described [33].

### 4.4. Isothermal Titration Calorimetry

Isothermal titration calorimetry (ITC) was used to determine the ligand-binding constant (K_D_) for binding to NR4A1 utilizing an Affinity ITC (TA Instruments, New Castle, DE, USA). Briefly, the experimental setup was as follows. The ITC sample cell contained 250 µL or NR4A1 protein (ligand-binding domain: LBD) at a concentration of 20 µmol/L in a buffer containing 20 mmol sodium phosphate/L (pH 7.4), 5% glycerol, and 1.0% ethanol. The ligand titrant was prepared in the same budder as that above at a ligand concentration of 100 µmol/L. The initial ligand stock solution was prepared at a final concentration of 10 mmol ligand/L ethanol prior to the preparation of the ligand titrant. Ligand titration into protein was performed at 25 °C with a stir rate of 125 rpm. Each ligand injection volume was 5 µL followed by a duration of 200 s to measure the total heat flow required to maintain constant temperature. A total of thirty injections were carried out for each ligand/NR4A1 combination. In a separate set of injections, the same ligand dilution was placed into the buffer. The ligand/buffer values were subtracted from the ligand/protein values prior to data analysis using the Affinity ITC manufacturer-supplied data analysis software package. The resulting data are plotted as heat flow (µJ) versus the molar ratio of the injected ligand to NR4A1 in the sample cell.

### 4.5. Computation-Based Molecular Modeling Studies

Molecular modeling studies were conducted using Maestro (Schrödinger Release 2021-3, Schrödinger, LLC, New York, NY, USA, 2021). The version of Maestro used for these studies was licensed to the Laboratory for Molecular Simulation (LMS), a Texas A&M University core-user facility for molecular modeling which is associated with the Texas A&M University’s High Performance Research Computing (HPRC) facility (College Station, TX 77843, USA). All Maestro-associated applications were accessed via the graphical user interface (GUI)’s VNC interactive application through the HPRC OnDemand portal. The crystal structure coordinates for the human orphan nuclear receptor NR4A1 ligand-binding domain (LBD) were downloaded from Protein Data Bank (http://www.fcsb.org, accessed on 26 April 2023. PDB ID 3V3Q). The human NR4A1 LBD crystal structure was prepared for ligand docking utilizing Maestro Protein Preparation Wizard; restrained minimization of the protein structure was performed utilizing the OPLS4 force field. The three-dimensional structure of each ligand was prepared for docking utilizing Maestro LigPrep. Maestro Glide [61,62,63] was utilized with the default settings to dock each prepared ligand to each prepared protein, predict the lowest energy ligand-binding orientation, and calculate the predicted binding energy in units of kcal/mol.

## Figures and Tables

**Figure 1 ijms-24-08152-f001:**
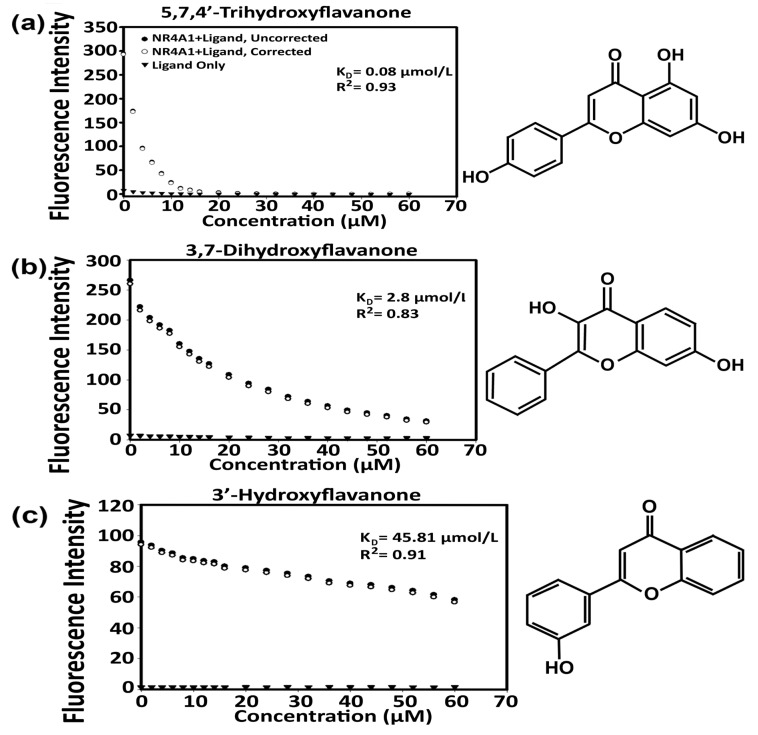
Binding of hydroxyflavones to the LBD of NR4A1: fluorescence assay. 5,7,4′-Trihydroxyflavone (**a**), 3,7-dihydroxyflavone (**b**), and 3′-hydroxyflavone (**c**) were incubated with the LBD of NR4A1, binding was determined using a fluorescent binding assay and K_D_ and R^2^ values were also determined as described [33].

**Figure 2 ijms-24-08152-f002:**
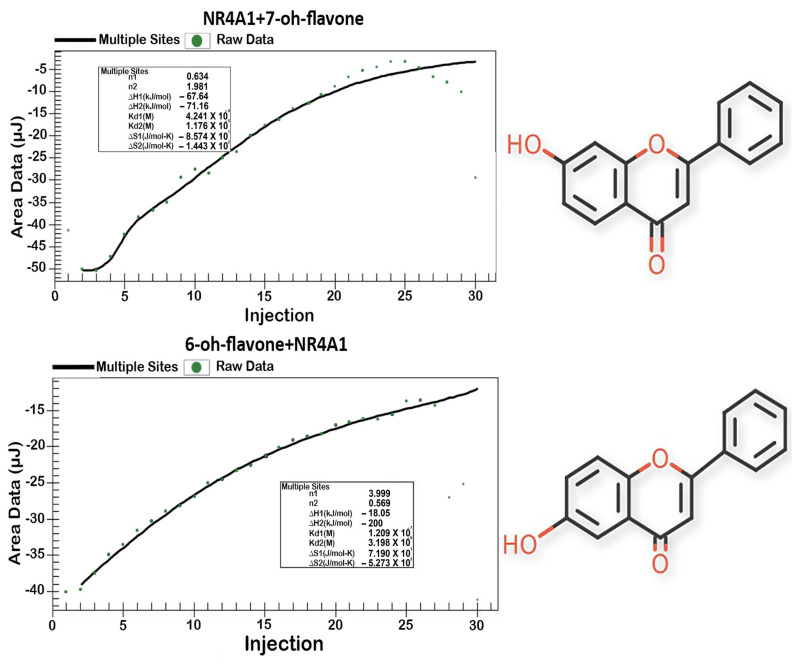
Binding of hydroxyflavones to the LBD of NR4A1: ITC assay. Interactions of 6-hydroxyflavone and 7-hydroxyflavone with the LBD of NR4A1 were determined using an Affinity ITC and analysis of the K_D_, and thermodynamic data were obtained as described using a data analysis software package supplied by the manufacturer.

**Figure 3 ijms-24-08152-f003:**
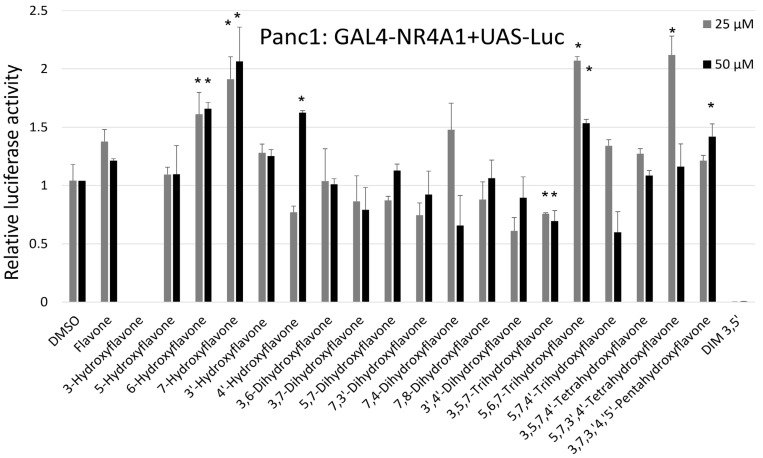
Hydroxyflavonoid activation of NR4A1-dependent transactivation. Panc1 cells were cotransfected with GAL4-NR4A1 and UAS-luciferase constructs, cells were treated with 25 and 50 µM hydroxyflavonoids and luciferase activity (normalized to β-galactosidase activity) was determined. Results were determined in triplicate for each concentration and are plotted as mean values ±SE. Significant induction or inhibition (*p* < 0.05) of luciferase activity is illustrated (*).

**Figure 4 ijms-24-08152-f004:**
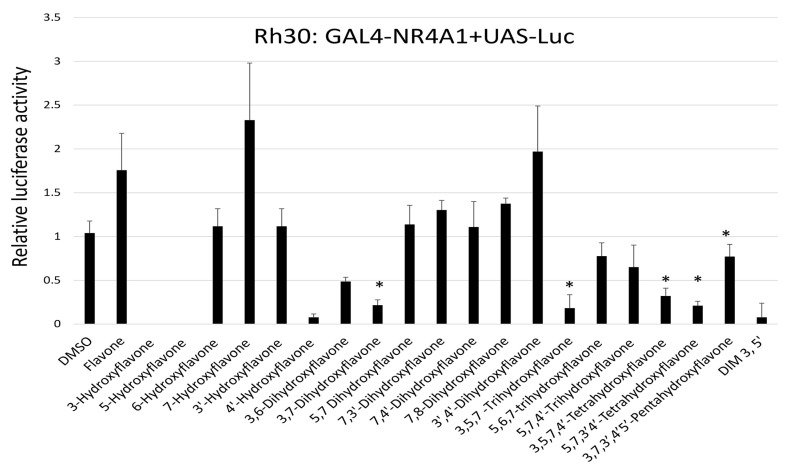
Hydroxyflavonoid activation of NR4A1-dependent transactivation. Rh30 cells were cotransfected with GAL4-NR4A1 and UAS-luciferase constructs, cells were treated with 50 µM hydroxyflavones and luciferase activity (normalized to β-galactosidase activity) was determined. Results were determined in triplicate for each concentration and are plotted as mean values ±SE. Significant induction or inhibition (*p* < 0.05) of luciferase activity is illustrated (*).

**Figure 5 ijms-24-08152-f005:**
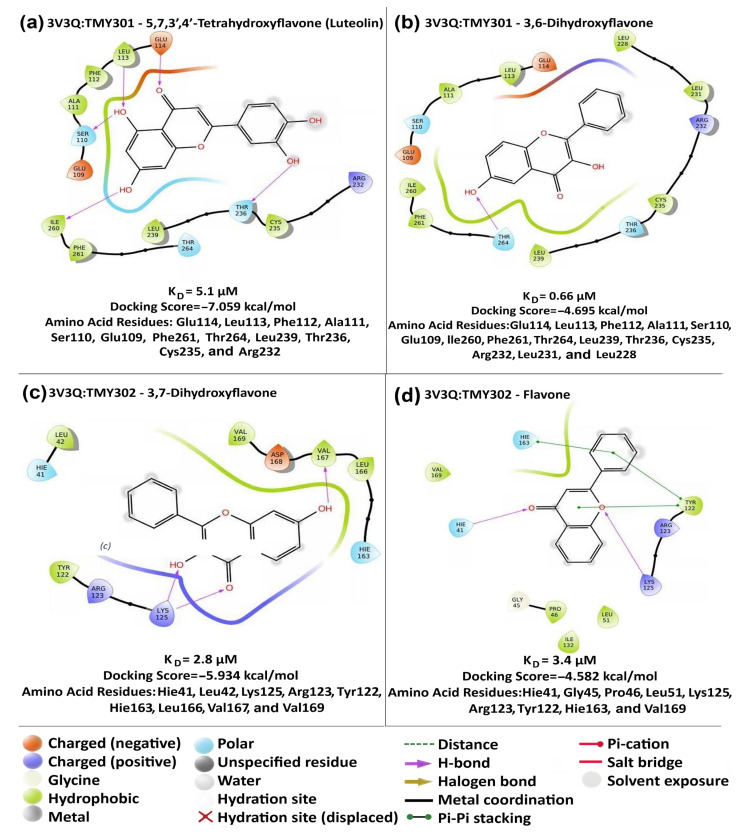
Modeling of hydroxyflavone binding to NR4A1. Modeling studies were carried out as described in the Methods and interactions of 5,7,3′,4′-tetrahydroxyflavone (**a**) and 3,6-dihydroxyflavone (**b**) gave docking scores of −5.1 and −4.695, respectively to TMY301. Interactions with amino acids are indicated. The docking scores of 3,7-dihydroxyflavone (**c**) and flavone (**d**) were −5.934 and −4.582, respectively to TMY302. The types of interactions include: hydrophobic (green), polar (light blue), positively charged (dark blue), negatively charged (red/orange), hydrogen bonding (purple).

**Table 1 ijms-24-08152-t001:** Binding K_D_ values of flavone, hydroxyflavones and naringenin to the ligand-binding domain of NR4A1.

Compound	Direct Binding Assay		ITC Assay	
	K_D_	R^2^	K_d_1, mmol/L	ΔG, kJ/mol
Flavone	3.4	0.9	0.075	−40.7
3-Hydroxyflavone	N/A *	N/A	0.0225	−43.6
5-Hydroxyflavone	1.4	0.87	18	−27
6-Hydroxyflavone	12.5	0.86	0.032	−42.9
7-Hydroxyflavone	7.4	0.93	0.042	−42.1
3′-Hydroxyflavone	45.8	0.91	0.33	−37
4′-Hydroxyflavone	13.02	0.9	0.15	−39.4
3,6-Dihydroxyflavone	0.66	0.98	444	−19.1
3,7-Dihydroxyflavone	2.8	0.83	0.18	−38.5
5,7 Dihydroxyflavone	11.3	0.89	0.001	−51.3
7,3′-Dihydroxyflavone	21.4	0.89	0.17	−38.7
7,4’-Dihydroxyflavone	9.1	0.68	0.49	−36.1
7,8-Dihydroxyflavone	17.64	0.99	0.042	−42.1
3′,4′-Dihydroxyflavone	0.96	0.59	9.5	−28.7
3,5,7 -Trihydroxyflavone (Galangin)	0.36	0.64	0.001	−51.2
5,6,7 -Trihydroxyflavone (Baicalein)	1.85	0.91	0.13	−39.4
5,7,4′-Trihydroxyflavone (Apigenin)	1.77	0.87	0.57	−35.7
3,5,7,4′-Tetrahydroxyflavone (Kaempferol)	3.1	0.95	97.1	−22.9
5,7,3′4′-Tetrahydroxyflavone (Luteolin)	5.1	0.98	0.019	−44.1
3,7,3′,4′5′-Pentahydroxyflavone (Quercetin)	0.81	0.97	0.35	−36.8

* 3-Hydroxyflavone is insoluble. (N/A)

**Table 2 ijms-24-08152-t002:** Summary of docking scores for flavone and hydroxyflavones from modeling studies.

Compound	Site TMY301: Docking Score kcal/mol	Site TMY302: Docking Score kcal/mol
Flavone	−5.741	−4.582
3-Hydroxyflavone	−5.507	−4.716
5-Hydroxyflavone	−6.302	−4.732
6-Hydroxyflavone	−5.619	−5.4
7-Hydroxyflavone	−5.878	−5.477
3′-Hydroxyflavone	−6.756	−5.176
4′-Hydroxyflavone	−5.861	−4.717
3,6-Dihydroxyflavone	−4.695	−5.51
3,7-Dihydroxyflavone	−5.934	−5.851
5,7 Dihydroxyflavone	−6.399	−5.543
7,3′-Dihydroxyflavone	−6.719	−5.559
7,4′-Dihydroxyflavone	−5.575	−5.42
7,8-Dihydroxyflavone	−5.941	−5.049
3′,4′-Dihydroxyflavone	−5.736	−5.507
3,5,7-Trihydroxyflavone (Galangin)	−6.791	−4.83
5,6,7-Trihydroxyflavone (Baicalein)	−5.711	−5.711
5,7,4′-Trihydroxyflavone (Apigenin)	−6.441	−4.782
3,5,7,4′-Tetrahydroxyflavone (Kaempferol)	−6.555	−5.433
5,7,3′4′-Tetrahydroxyflavone (Luteolin)	−7.059	−5.832
3,7,3′,4′5′-Pentahydroxyflavone (Quercetin)	−6.437	−5.397
4′,5,7-Trihydroxyflavone (Naringenin)	−6.618	−5.656
Binding Pocket Residues	GLU114, LEU113, PHE112, ALA111, SER110, GLU109, LEU108, ILE260, PHE261, THR264, PRO266, LEU239, THR236, CYS235, ARG232, ARG184, VAL179	HIE41, LEU42, ASP43, SER44, GLY45, PRO46, SER47, THR48, LEU51, ILE132, LYS125, ARG123, TYR122, ARG119, LEU162, HIE163, LEU165, LEU166, VAL167, VAL169, PHE172

## Data Availability

The data that support the findings of this study are available from the corresponding author upon reasonable request.

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
