# Peer review of "Flavone and Hydroxyflavones Are Ligands That Bind the Orphan Nuclear Receptor 4A1 (NR4A1)"

_ijms, 2023, doi:10.3390/ijms24098152_

Round 1

Reviewer 1 Report

The article overall is well written, however there are some minor changes throughout the text to be improved.

There is inconsistency when naming the compounds (6 hydroxyflavones, 7 dihydroxyflavones, 3 trihydroxyflavones, 2 tetrahydroxyflavones, and 1 pentahydroxyflavone) in the abstract the is no dash, throughout the text, different kind of dash style appear, somewhere there is space between number and dash, so please uniform it.

Figures 1, 2 the resolution of the image must be higher, see the author guidelines (https://www.mdpi.com/journal/ijms/instructions) for resolution of the figures. Also, text on the graphs on the left side of the image should be a little bigger as it is barely readable.

Figure 5 also needs improvement in resolution and font size

References are not in adequate format (https://www.mdpi.com/journal/ijms/instructions#references)

Author Response

Thank you for taking the time to review and critic our manuscript. Below please find our responses to your concerns.

Comments and Suggestions for Authors

The article overall is well written, however there are some minor changes throughout the text to be improved.

There is inconsistency when naming the compounds (6 hydroxyflavones, 7 dihydroxyflavones, 3 trihydroxyflavones, 2 tetrahydroxyflavones, and 1 pentahydroxyflavone) in the abstract the is no dash, throughout the text, different kind of dash style appear, somewhere there is space between number and dash, so please uniform it.

  • Author Response: The inconsistencies have now been corrected.

Figures 1, 2 the resolution of the image must be higher, see the author guidelines (https://www.mdpi.com/journal/ijms/instructions) for resolution of the figures. Also, text on the graphs on the left side of the image should be a little bigger as it is barely readable.

Figure 5 also needs improvement in resolution and font size

  • Author Response: The figures have now been improved and are more readable.

References are not in adequate format (https://www.mdpi.com/journal/ijms/instructions#references)

  • Author Response: The reference format has now been corrected.

Reviewer 2 Report

The manuscript by Stephen Safe and co-workers reports that Flavone and Hydroxyflavones are ligands that bind the NR4A1. This manuscript clearly describes the findings and adds to the field of orphan nuclear receptor 4A1. The conclusion that Flavone and Hydroxyflavones are ligands of NR4A1 is intriguing and opens up new avenues for future research. However, some critical points need to be addressed to strengthen the claims made in the study.

  1. Figure 3 needs reevaluation as there appear to be some inconsistencies in the plots and their accompanying description. The data bar and error bar for 5-Hydroxyflavone, 6-Hydroxyflavone does not display any difference where it has been reported to be significantly induced or inhibited.
  2. The visibility of Figure 1 data plot and Figure 5 are generally poor. They may be improved by dedicated software.                                                                     In conclusion, the study is a good starting point, but these points should be addressed in order to improve the manuscript.

Author Response

Thank you for taking the time to review and critic our manuscript. Below please find our responses to your concerns.

Comments and Suggestions for Authors

The manuscript by Stephen Safe and co-workers reports that Flavone and Hydroxyflavones are ligands that bind the NR4A1. This manuscript clearly describes the findings and adds to the field of orphan nuclear receptor 4A1. The conclusion that Flavone and Hydroxyflavones are ligands of NR4A1 is intriguing and opens up new avenues for future research. However, some critical points need to be addressed to strengthen the claims made in the study.

Figure 3 needs reevaluation as there appear to be some inconsistencies in the plots and their accompanying description. The data bar and error bar for 5-Hydroxyflavone, 6-Hydroxyflavone does not display any difference where it has been reported to be significantly induced or inhibited.

  • Author Response: The description of inducible luciferase activity in the text now match induction responses in Figure 3.

The visibility of Figure 1 data plot and Figure 5 are generally poor. They may be improved by dedicated software.

Author Response: Figures 1 and 5 have now been improved.